# Convergence of Cubic Regularization for Nonconvex Optimization under KŁ Property

**Yi Zhou**
Department of ECE
The Ohio State University
zhou.1172@osu.edu

**Zhe Wang**
Department of ECE
The Ohio State University
wang.10982@osu.edu

**Yingbin Liang**
Department of ECE
The Ohio State University
liang.889@osu.edu

## Abstract

Cubic-regularized Newton's method (CR) is a popular algorithm that guarantees to produce a second-order stationary solution for solving nonconvex optimization problems. However, existing understandings of the convergence rate of CR are conditioned on special types of geometrical properties of the objective function. In this paper, we explore the asymptotic convergence rate of CR by exploiting the ubiquitous Kurdyka-Łojasiewicz (KŁ) property of nonconvex objective functions. In specific, we characterize the asymptotic convergence rate of various types of optimality measures for CR including function value gap, variable distance gap, gradient norm and least eigenvalue of the Hessian matrix. Our results fully characterize the diverse convergence behaviors of these optimality measures in the full parameter regime of the KŁ property. Moreover, we show that the obtained asymptotic convergence rates of CR are order-wise faster than those of first-order gradient descent algorithms under the KŁ property.

## 1 Introduction

A majority of machine learning applications are naturally formulated as nonconvex optimization due to the complex mechanism of the underlying model. Typical examples include training neural networks in deep learning Goodfellow et al. (2016), low-rank matrix factorization Ge et al. (2016); Bhojanapalli et al. (2016), phase retrieval Candès et al. (2015); Zhang et al. (2017), etc. In particular, these machine learning problems take the generic form

$$\min_{\mathbf{x} \in \mathbb{R}^d} \quad f(\mathbf{x}), \tag{P}$$

where the objective function $f : \mathbb{R}^d \to \mathbb{R}$ is a differentiable and nonconvex function, and usually corresponds to the total loss over a set of data samples. Traditional first-order algorithms such as gradient descent can produce a solution $\bar{\mathbf{x}}$ that satisfies the first-order stationary condition, i.e., $\nabla f(\bar{\mathbf{x}}) = \mathbf{0}$. However, such first-order stationary condition does not exclude the possibility of approaching a saddle point, which can be a highly suboptimal solution and therefore deteriorate the performance.

Recently in the machine learning community, there has been an emerging interest in designing algorithms to scape saddle points in nonconvex optimization, and one popular algorithm is the cubic-regularized Newton's algorithm Nesterov and Polyak (2006); Agarwal et al. (2017); Yue et al.

(2018), which is also referred to as cubic regularization (CR) for simplicity. Such a second-order method exploits the Hessian information and produces a solution $\bar{\mathbf{x}}$ for problem (P) that satisfies the second-order stationary condition, i.e.,

$$\text{(second-order stationary)}: \quad \nabla f(\bar{\mathbf{x}}) = \mathbf{0}, \ \nabla^2 f(\bar{\mathbf{x}}) \succeq \mathbf{0}.$$

The second-order stationary condition ensures CR to escape saddle points wherever the corresponding Hessian matrix has a negative eigenvalue (i.e., strict saddle points Sun et al. (2015)). In particular, many nonconvex machine learning problems have been shown to exclude spurious local minima and have only strict saddle points other than global minima Baldi and Hornik (1989); Sun et al. (2017); Ge et al. (2016); Zhou and Liang (2018). In such a desirable case, CR is guaranteed to find the global minimum for these nonconvex problems. To be specific, given a proper parameter $M > 0$, CR generates a variable sequence $\{\mathbf{x}_k\}_k$ via the following update rule

$$\text{(CR) } \mathbf{x}_{k+1} \in \underset{\mathbf{y} \in \mathbb{R}^d}{\operatorname{argmin}} \ \langle \mathbf{y} - \mathbf{x}_k, \nabla f(\mathbf{x}_k) \rangle + \frac{1}{2} (\mathbf{y} - \mathbf{x}_k)^\top \nabla^2 f(\mathbf{x}_k)(\mathbf{y} - \mathbf{x}_k) + \frac{M}{6} \|\mathbf{y} - \mathbf{x}_k\|^3. \quad (1)$$

Intuitively, CR updates the variable by minimizing an approximation of the objective function at the current iterate. This approximation is essentially the third-order Taylor's expansion of the objective function. We note that computationally efficient solver has been proposed for solving the above cubic subproblem Agarwal et al. (2017), and it was shown that the resulting computation complexity of CR to achieve a second-order stationary point serves as the state-of-the-art among existing algorithms that achieve the second-order stationary condition.

Several studies have explored the convergence of CR for nonconvex optimization. Specifically, the pioneering work Nesterov and Polyak (2006) first proposed the CR algorithm and showed that the variable sequence $\{\mathbf{x}_k\}_k$ generated by CR has second-order stationary limit points and converges sub-linearly. Furthermore, the gradient norm along the iterates was shown to converge quadratically (i.e., super-linearly) given an initialization with positive definite Hessian. Moreover, the function value along the iterates was shown to converge super-linearly under a certain gradient dominance condition of the objective function. Recently, Yue et al. (2018) studied the asymptotic convergence rate of CR under the local error bound condition, and established the quadratic convergence of the distance between the variable and the solution set. Clearly, these results demonstrate that special geometrical properties (such as the gradient dominance condition and the local error bound) enable much faster convergence rate of CR towards a second-order stationary point. However, these special conditions may fail to hold for generic complex objectives involved in practical applications. Thus, it is desired to further understand the convergence behaviors of CR for optimizing nonconvex functions over a broader spectrum of geometrical conditions.

The Kurdyka-Łojasiewicz (KŁ) property (see Section 2 for details) Bolte et al. (2007, 2014) serves as such a candidate. As described in Section 2, the KŁ property captures a broad spectrum of the local geometries that a nonconvex function can have and is parameterized by a parameter $\theta$ that changes over its allowable range. In fact, the KŁ property has been shown to hold ubiquitously for most practical functions (see Section 2 for a list of examples). The KŁ property has been exploited extensively to analyze the convergence rate of various *first-order* algorithms for nonconvex optimization, e.g., gradient method Attouch and Bolte (2009); Li et al. (2017), alternating minimization Bolte et al. (2014) and distributed gradient methods Zhou et al. (2016a). But it has not been exploited to establish the *convergence rate* of second-order algorithms towards second-order stationary points. In this paper, we exploit the KŁ property of the objective function to provide a comprehensive study of the convergence rate of CR for nonconvex optimization. We anticipate our study to substantially advance the existing understanding of the convergence of CR to a much broader range of nonconvex functions. We summarize our contributions as follows.

## 1.1 Our Contributions

We characterize the convergence rate of CR locally towards second-stationary points for nonconvex optimization under the KŁ property of the objective function. Our results also compare and contrast the different order-level of guarantees that KŁ yields between the first and second-order algorithms. Also, our analysis characterizes the convergence property for a range of KŁ parameter $\theta$, which requires the development of different recursive inequalities for various cases.

**Gradient norm & least eigenvalue of Hessian:** We characterize the asymptotic convergence rates of the sequence of gradient norm $\{\|\nabla f(\mathbf{x}_k)\|\}_k$ and the sequence of the least eigenvalue of Hessian

$\{\lambda_{\min}(\nabla^2 f(\mathbf{x}_k))\}_k$ generated by CR. We show that CR meets the second-order stationary condition at a (super)-linear rate in the parameter range $\theta \in [\frac{1}{3}, 1]$ for the KŁ property, while the convergence rates become sub-linearly in the parameter range $\theta \in (0, \frac{1}{3})$ for the KŁ property.

**Function value gap & variable distance:** We characterize the asymptotic convergence rates of the function value sequence $\{f(\mathbf{x}_k)\}_k$ and the variable sequence $\{\mathbf{x}_k\}_k$ to the function value and the variable of a second-order stationary point, respectively. The obtained convergence rates range from sub-linearly to super-linearly depending on the parameter $\theta$ associated with the KŁ property. Our convergence results generalize the existing ones established in Nesterov and Polyak (2006) corresponding to special cases of $\theta$ to the full range of $\theta$. Furthermore, these types of convergence rates for CR are orderwise faster than the corresponding convergence rates of the first-order gradient descent algorithm in various regimes of the KŁ parameter $\theta$ (see Table 1 and Table 2 for the comparisons).

Based on the KŁ property, we establish a generalized local error bound condition (referred to as the KŁ error bound). The KŁ error bound further leads to characterization of the convergence rate of distance between the variable and the solution set, which ranges from sub-linearly to super-linearly depending on the parameter $\theta$ associated with the KŁ property. This result generalizes the existing one established in Yue et al. (2018) corresponding to a special case of $\theta$ to the full range of $\theta$.

## 1.2 Related Work

**Cubic regularization:** CR algorithm was first proposed in Nesterov and Polyak (2006), in which the authors analyzed the convergence of CR to second-order stationary points for nonconvex optimization. In Nesterov (2008), the authors established the sub-linear convergence of CR for solving convex smooth problems, and they further proposed an accelerated version of CR with improved sub-linear convergence. Recently, Yue et al. (2018) studied the asymptotic convergence properties of CR under the error bound condition, and established the quadratic convergence of the iterates. Several other works proposed different methods to solve the cubic subproblem of CR, e.g., Agarwal et al. (2017); Carmon and Duchi (2016); Cartis et al. (2011b).

**Inexact-cubic regularization:** Another line of work aimed at improving the computation efficiency of CR by solving the cubic subproblem with inexact gradient and Hessian information. In particular, Ghadimi et al. (2017); Tripuraneni et al. (2017) proposed an inexact CR for solving convex problem and nonconvex problems, respectively. Also, Cartis et al. (2011a) proposed an adaptive inexact CR for nonconvex optimization, whereas Jiang et al. (2017) further studied the accelerated version for convex optimization. Several studies explored subsampling schemes to implement inexact CR algorithms, e.g., Kohler and Lucchi (2017); Xu et al. (2017); **?**); Wang et al. (2018).

**KŁ property:** The KŁ property was first established in Bolte et al. (2007), and was then widely applied to characterize the asymptotic convergence behavior for various first-order algorithms for nonconvex optimization Attouch and Bolte (2009); Li et al. (2017); Bolte et al. (2014); Zhou et al. (2016a); Zhou et al. (2018). The KŁ property was also exploited to study the convergence of second-order algorithms such as generalized Newton's method Frankel et al. (2015) and the trust region method Noll and Rondepierre (2013). However, these studies did not characterize the convergence rate and the studied methods cannot guarantee to converge to second-order stationary points, whereas this paper provides this type of results.

**First-order algorithms that escape saddle points:** Many first-order algorithms are proved to achieve second-order stationary points for nonconvex optimization. For example, online stochastic gradient descent Ge et al. (2015), perturbed gradient descent Tripuraneni et al. (2017), gradient descent with negative curvature Carmon and Duchi (2016); Liu and Yang (2017) and other stochastic algorithms Allen-Zhu (2017).

## 2 Preliminaries on KŁ Property and CR Algorithm

Throughout the paper, we make the following standard assumptions on the objective function $f : \mathbb{R}^d \to \mathbb{R}$ Nesterov and Polyak (2006); Yue et al. (2018).

**Assumption 1.** *The objective function $f$ in the problem* (P) *satisfies:*

*1. Function $f$ is continuously twice-differentiable and bounded below, i.e., $\inf_{\mathbf{x} \in \mathbb{R}^d} f(\mathbf{x}) > -\infty$;*
*2. For any $\alpha \in \mathbb{R}$, the sub-level set $\{\mathbf{x} : f(\mathbf{x}) \le \alpha\}$ is compact;*

*3. The Hessian of $f$ is $L$-Lipschitz continuous on a compact set $\mathcal{C}$, i.e.,*

$$\|\nabla^2 f(\mathbf{x}) - \nabla^2 f(\mathbf{y})\| \le L\|\mathbf{x} - \mathbf{y}\|, \quad \mathbf{x}, \mathbf{y} \in \mathcal{C}.$$

The above assumptions make problem (P) have a solution and make the iterative rule of CR being well defined. We note that if the Lipschitz constant is unknown a priori, one can apply a standard back-tracking line search step to dynamically search for a proper constant. Besides these assumptions, many practical functions are shown to satisfy the so-called Łojasiewicz gradient inequality Łojasiewicz (1965). Recently, such a condition was further generalized to the Kurdyka-Łojasiewicz property Bolte et al. (2007, 2014), which is satisfied by a larger class of nonconvex functions.

Next, we introduce the KŁ property of a function $f$. Throughout, the (limiting) subdifferential of a proper and lower-semicontinuous function $f$ is denoted as $\partial f$, and the point-to-set distance is denoted as $\mathrm{dist}_\Omega(\mathbf{x}) := \inf_{\mathbf{w} \in \Omega} \|\mathbf{x} - \mathbf{w}\|$.

**Definition 1** (KŁ property, Bolte et al. (2014)). *A proper and lower-semicontinuous function $f$ is said to satisfy the KŁ property if for every compact set $\Omega \subset \mathrm{dom} f$ on which $f$ takes a constant value $f_\Omega \in \mathbb{R}$, there exist $\varepsilon, \lambda > 0$ such that for all $\bar{\mathbf{x}} \in \Omega$ and all $\mathbf{x} \in \{\mathbf{z} \in \mathbb{R}^d : \mathrm{dist}_\Omega(\mathbf{z}) < \varepsilon, f_\Omega < f(\mathbf{z}) < f_\Omega + \lambda\}$, one has*

$$\varphi'\left(f(\mathbf{x}) - f_\Omega\right) \cdot \mathrm{dist}_{\partial f(\mathbf{x})}(\mathbf{0}) \ge 1, \tag{2}$$

*where $\varphi'$ is the derivative of function $\varphi : [0, \lambda) \to \mathbb{R}_+$, which takes the form $\varphi(t) = \frac{c}{\theta} t^\theta$ for some constants $c > 0, \theta \in (0, 1]$.*

The KŁ property establishes local geometry of the nonconvex function around a compact set. In particular, consider a differentiable function (which is our interest here) so that $\partial f = \nabla f$. Then, the local geometry described in eq. (2) can be rewritten as

$$f(\mathbf{x}) - f_\Omega \le C\|\nabla f(\mathbf{x})\|^{\frac{1}{1-\theta}} \tag{3}$$

for some constant $C > 0$. Equation (3) can be viewed as a generalization of the gradient dominance condition Łojasiewicz (1963); Karimi et al. (2016), which corresponds to the special case of $\theta = \frac{1}{2}$. The KŁ property has been shown to hold for a large class of functions including sub-analytic functions, logarithm and exponential functions and semi-algebraic functions. These function classes cover most of nonconvex objective functions encountered in practical applications. To illustrate, consider the function $x^{p/q}$, where $p$ is an even positive integer and $q < p$ is a positive integer. Then, it satisfies the KŁ property in the region $[-1, 1]$ with $\theta = 1 - \frac{p-q}{p}$. More generally, many nonconvex problems have been shown to be KŁ with $\theta = 1/2$ Zhou et al. (2016b); Yue et al. (2018); Zhou and Liang (2017).

Next, we provide some fundamental understandings of CR that determines the convergence rate. The algorithmic dynamics of CR Nesterov and Polyak (2006) is very different from that of first-order gradient descent algorithm, which implies that the convergence rate of CR under the KŁ property can be very different from the existing result for the first-order algorithms under the KŁ property. We provide a detailed comparison between the two algorithmic dynamics in Table 1 for illustration, where $L_{\mathrm{grad}}$ corresponds to the Lipschitz parameter of $\nabla f$ for gradient descent, $L$ is the Lipschitz parameter for Hessian and we choose $M = L$ for CR for simple illustration.

Table 1: Comparison between dynamics of gradient descent and CR.

| | | gradient descent | cubic-regularization ($M = L$) |
|---|---|---|---|
| $f(\mathbf{x}_k) - f(\mathbf{x}_{k-1})$ | $\le$ | $-\frac{L_{\mathrm{grad}}}{2}\|\mathbf{x}_k - \mathbf{x}_{k-1}\|^2$ | $-\frac{L}{12}\|\mathbf{x}_k - \mathbf{x}_{k-1}\|^3$ |
| $\|\nabla f(\mathbf{x}_k)\|$ | $\le$ | $L_{\mathrm{grad}}\|\mathbf{x}_k - \mathbf{x}_{k-1}\|$ | $L\|\mathbf{x}_k - \mathbf{x}_{k-1}\|^2$ |
| $-\lambda_{\min}(\nabla^2 f(\mathbf{x}_k))$ | $\le$ | N/A | $\frac{3L}{2}\|\mathbf{x}_k - \mathbf{x}_{k-1}\|$ |

It can be seen from Table 1 that the dynamics of gradient descent involves information up to the first order (i.e., function value and gradient), whereas the dynamics of CR involves the additional second order information, i.e., least eigenvalue of Hessian (last line in Table 1). In particular, note that the successive difference of function value $f(\mathbf{x}_{k+1}) - f(\mathbf{x}_k)$ and the gradient norm $\|\nabla f(\mathbf{x}_k)\|$ of CR are bounded by higher order terms of $\|\mathbf{x}_k - \mathbf{x}_{k-1}\|$ compared to those of gradient descent. Intuitively,

this implies that CR should converge faster than gradient descent in the converging phase when $\mathbf{x}_{k+1} - \mathbf{x}_k \to \mathbf{0}$. Next, we exploit the dynamics of CR and the KŁ property to study its asymptotic convergence rate.

**Notation:** Throughout the paper, we denote $f(n) = \Theta(g(n))$ if and only if for some $0 < c_1 < c_2$, $c_1 g(n) \le f(n) \le c_2 g(n)$ for all $n \ge n_0$.

## 3 Convergence Rate of CR to Second-order Stationary Condition

In this subsection, we explore the convergence rates of the gradient norm and the least eigenvalue of the Hessian along the iterates generated by CR under the KŁ property. Define the second-order stationary gap

$$\mu(\mathbf{x}) := \max\left\{ \sqrt{\frac{2}{L+M}\|\nabla f(\mathbf{x})\|}, \; -\frac{2}{2L+M}\lambda_{\min}(\nabla^2 f(\mathbf{x})) \right\}.$$

The above quantity is well established as a criterion for achieving second-order stationary Nesterov and Polyak (2006). It tracks both the gradient norm and the least eigenvalue of the Hessian at $\mathbf{x}$. In particular, the second-order stationary condition is satisfied as $\mu(\mathbf{x}) = 0$.

Next, we characterize the convergence rate of $\mu$ for CR under the KŁ property.

**Theorem 1.** *Let Assumption 1 hold and assume that problem* $(\mathrm{P})$ *satisfies the KŁ property associated with parameter* $\theta \in (0, 1]$. *Then, there exists a sufficiently large* $k_0 \in \mathbb{N}$ *such that for all* $k \ge k_0$ *the sequence* $\{\mu(\mathbf{x}_k)\}_k$ *generated by CR satisfies*

1. *If* $\theta = 1$, *then* $\mu(\mathbf{x}_k) \to 0$ *within finite number of iterations;*
2. *If* $\theta \in (\frac{1}{3}, 1)$, *then* $\mu(\mathbf{x}_k) \to 0$ *super-linearly as* $\mu(\mathbf{x}_k) \le \Theta\left( \exp\left( -\left(\frac{2\theta}{1-\theta}\right)^{k-k_0} \right) \right)$;
3. *If* $\theta = \frac{1}{3}$, *then* $\mu(\mathbf{x}_k) \to 0$ *linearly as* $\mu(\mathbf{x}_k) \le \Theta\left( \exp\left( -(k - k_0) \right) \right)$;
4. *If* $\theta \in (0, \frac{1}{3})$, *then* $\mu(\mathbf{x}_k) \to 0$ *sub-linearly as* $\mu(\mathbf{x}_k) \le \Theta\left( (k-k_0)^{-\frac{2\theta}{1-3\theta}} \right)$.

Theorem 1 provides a full characterization of the convergence rate of $\mu$ for CR to meet the second-order stationary condition under the KŁ property. It can be seen from Theorem 1 that the convergence rate of $\mu$ is determined by the KŁ parameter $\theta$. Intuitively, in the regime $\theta \in (0, \frac{1}{3}]$ where the local geometry is 'flat', CR achieves second-order stationary slowly as (sub)-linearly. As a comparison, in the regime $\theta \in (\frac{1}{3}, 1]$ where the local geometry is 'sharp', CR achieves second-order stationary fast as super-linearly.

We next compare the convergence results of $\mu$ in Theorem 1 with that of $\mu$ for CR studied in Nesterov and Polyak (2006). To be specific, the following two results are established in (Nesterov and Polyak, 2006, Theorems 1 & 3) under Assumption 1.

1. $\lim_{k \to \infty} \mu(\mathbf{x}_k) = 0, \quad \min_{1 \le t \le k} \mu(\mathbf{x}_t) \le \Theta(k^{-\frac{1}{3}})$;
2. If the Hessian is positive definite at certain $\mathbf{x}_t$, then the Hessian remains to be positive definite at all subsequent iterates $\{\mathbf{x}_k\}_{k \ge t}$ and $\{\mu(\mathbf{x}_k)\}_k$ converges to zero quadratically.

Item 1 establishes a best-case bound for $\mu(\mathbf{x}_k)$, i.e., it holds only for the minimum $\mu$ along the iteration path. As a comparison, our results in Theorem 1 characterize the convergence rate of $\mu(\mathbf{x}_k)$ along the entire asymptotic iteration path. Also, the quadratic convergence in item 2 relies on the fact that CR eventually stays in a locally strong convex region (with KŁ parameter $\theta = \frac{1}{2}$), and this is consistent with our convergence rate in Theorem 1 for KŁ parameter $\theta = \frac{1}{2}$. In summary, our result captures the effect of the underlying geometry (parameterized by the KŁ parameter $\theta$) on the convergence rate of CR towards second-order stationary.

## 4 Other Convergence Results of CR

In this section, we first present the convergence rate of function value and variable distance for CR under the KŁ property. Then, we discuss that such convergence results are also applicable to characterize the convergence rate of inexact CR.

## 4.1 Convergence Rate of Function Value for CR

It has been proved in (Nesterov and Polyak, 2006, Theorem 2) that the function value sequence $\{f(\mathbf{x}_k)\}_k$ generated by CR decreases to a finite limit $\bar{f}$, which corresponds to the function value evaluated at a certain second-order stationary point. The corresponding convergence rate has also been developed in Nesterov and Polyak (2006) under certain gradient dominance condition of the objective function. In this section, we characterize the convergence rate of $\{f(\mathbf{x}_k)\}_k$ to $\bar{f}$ for CR by exploiting the more general KŁ property. We obtain the following result.

**Theorem 2.** *Let Assumption 1 hold and assume problem* (P) *satisfies the KŁ property associated with parameter* $\theta \in (0, 1]$. *Then, there exists a sufficiently large* $k_0 \in \mathbb{N}$ *such that for all* $k \geq k_0$ *the sequence* $\{f(\mathbf{x}_k)\}_k$ *generated by CR satisfies*

*1. If* $\theta = 1$, *then* $f(\mathbf{x}_k) \downarrow \bar{f}$ *within finite number of iterations;*

*2. If* $\theta \in (\frac{1}{3}, 1)$, *then* $f(\mathbf{x}_k) \downarrow \bar{f}$ *super-linearly as* $f(\mathbf{x}_{k+1}) - \bar{f} \leq \Theta\left(\exp\left(-\left(\frac{2}{3(1-\theta)}\right)^{k-k_0}\right)\right)$;

*3. If* $\theta = \frac{1}{3}$, *then* $f(\mathbf{x}_k) \downarrow \bar{f}$ *linearly as* $f(\mathbf{x}_{k+1}) - \bar{f} \leq \Theta\left(\exp\left(-(k - k_0)\right)\right)$;

*4. If* $\theta \in (0, \frac{1}{3})$, *then* $f(\mathbf{x}_k) \downarrow \bar{f}$ *sub-linearly as* $f(\mathbf{x}_{k+1}) - \bar{f} \leq \Theta\left((k - k_0)^{-\frac{2}{1-3\theta}}\right)$.

From Theorem 2, it can be seen that the function value sequence generated by CR has diverse asymptotic convergence rates in different regimes of the KŁ parameter $\theta$. In particular, a larger $\theta$ implies a sharper local geometry that further facilitates the convergence. We note that the gradient dominance condition discussed in Nesterov and Polyak (2006) locally corresponds to the KŁ property in eq. (3) with the special cases $\theta \in \{0, \frac{1}{2}\}$, and hence the convergence rate results in Theorem 2 generalize those in (Nesterov and Polyak, 2006, Theorems 6, 7).

We can further compare the function value convergence rates of CR with those of gradient descent method Frankel et al. (2015) under the KŁ property (see Table 2 for the comparison). In the KŁ parameter regime $\theta \in [\frac{1}{3}, 1]$, the convergence rate of $\{f(\mathbf{x}_k)\}_k$ for CR is super-linear—orderwise faster than the corresponding (sub)-linear convergence rate of gradient descent. Also, both methods converge sub-linearly in the parameter regime $\theta \in (0, \frac{1}{3})$, and the corresponding convergence rate of CR is still considerably faster than that of gradient descent.

Table 2: Comparison of convergence rate of $\{f(\mathbf{x}_k)\}_k$ between gradient descent and CR.

| KŁ parameter | gradient descent | cubic-regularization |
|:---:|:---:|:---:|
| $\theta = 1$ | finite-step | finite-step |
| $\theta \in [\frac{1}{2}, 1)$ | linear | super-linear |
| $\theta \in [\frac{1}{3}, \frac{1}{2})$ | sub-linear | (super)-linear |
| $\theta \in (0, \frac{1}{3})$ | sub-linear $\mathcal{O}(k^{-\frac{1}{1-2\theta}})$ | sub-linear $\mathcal{O}(k^{-\frac{2}{1-3\theta}})$ |

## 4.2 Convergence Rate of Variable Distance for CR

It has been proved in (Nesterov and Polyak, 2006, Theorem 2) that all limit points of $\{\mathbf{x}_k\}_k$ generated by CR are second order stationary points. However, the sequence is not guaranteed to be convergent and no convergence rate is established.

Our next two results show that the sequence $\{\mathbf{x}_k\}_k$ generated by CR is convergent under the KŁ property.

**Theorem 3.** *Let Assumption 1 hold and assume that problem* (P) *satisfies the KŁ property. Then, the sequence* $\{\mathbf{x}_k\}_k$ *generated by CR satisfies*

$$\sum_{k=0}^{\infty} \|\mathbf{x}_{k+1} - \mathbf{x}_k\| < +\infty. \tag{4}$$

Theorem 3 implies that CR generates an iterate $\{\mathbf{x}_k\}_k$ with finite trajectory length, i.e., $\|\mathbf{x}_\infty - \mathbf{x}_0\| < +\infty$. In particular, eq. (4) shows that the sequence is absolutely summable, which strengthens

the result in Nesterov and Polyak (2006) that establishes the cubic summability instead, i.e., $\sum_{k=0}^{\infty} \|\mathbf{x}_{k+1} - \mathbf{x}_k\|^3 < +\infty$. We note that the cubic summability property does not guarantee that the sequence is convergence, for example, the sequence $\{\frac{1}{k}\}_k$ is cubic summable but is not absolutely summable. In comparison, the summability property in eq. (4) directly implies that the sequence $\{\mathbf{x}_k\}_k$ generated by CR is a Cauchy convergent sequence, and we obtain the following result.

**Corollary 1.** *Let Assumption 1 hold and assume that problem* (P) *satisfies the KŁ property. Then, the sequence* $\{\mathbf{x}_k\}_k$ *generated by CR is a Cauchy sequence and converges to some second-order stationary point* $\bar{\mathbf{x}}$.

We note that the convergence of $\{\mathbf{x}_k\}_k$ to a second-order stationary point is also established for CR in Yue et al. (2018), but under the special error bound condition, whereas we establish the convergence of $\{\mathbf{x}_k\}_k$ under the KŁ property that holds for general nonconvex functions.

Next, we establish the convergence rate of $\{\mathbf{x}_k\}_k$ to the second-order stationary limit $\bar{\mathbf{x}}$.

**Theorem 4.** *Let Assumption 1 hold and assume that problem* (P) *satisfies the KŁ property. Then, there exists a sufficiently large* $k_0 \in \mathbb{N}$ *such that for all* $k \geq k_0$ *the sequence* $\{\mathbf{x}_k\}_k$ *generated by CR satisfies*

1. *If* $\theta = 1$, *then* $\mathbf{x}_k \to \bar{\mathbf{x}}$ *within finite number of iterations;*
2. *If* $\theta \in (\frac{1}{3}, 1)$, *then* $\mathbf{x}_k \to \bar{\mathbf{x}}$ *super-linearly as* $\|\mathbf{x}_{k+1} - \bar{\mathbf{x}}\| \leq \Theta\left( \exp\left( -\left(\frac{2\theta}{3(1-\theta)} + \frac{2}{3}\right)^{k-k_0}\right)\right);$
3. *If* $\theta = \frac{1}{3}$, *then* $\mathbf{x}_k \to \bar{\mathbf{x}}$ *linearly as* $\|\mathbf{x}_{k+1} - \bar{\mathbf{x}}\| \leq \Theta\left( \exp\left( -(k-k_0)\right)\right);$
4. *If* $\theta \in (0, \frac{1}{3})$, *then* $\mathbf{x}_k \to \bar{\mathbf{x}}$ *sub-linearly as* $\|\mathbf{x}_{k+1} - \bar{\mathbf{x}}\| \leq \Theta\left((k-k_0)^{-\frac{2\theta}{1-3\theta}}\right).$

From Theorem 4, it can be seen that the convergence rate of $\{\mathbf{x}_k\}_k$ is similar to that of $\{f(\mathbf{x}_k)\}_k$ in Theorem 2 in the corresponding regimes of the KŁ parameter $\theta$. Essentially, a larger parameter $\theta$ induces a sharper local geometry that leads to a faster convergence.

We can further compare the variable convergence rate of CR in Theorem 4 with that of gradient descent method Attouch and Bolte (2009) (see Table 3 for the comparison). It can be seen that the variable sequence generated by CR converges orderwise faster than that generated by the gradient descent method in a large parameter regimes of $\theta$.

Table 3: Comparison of convergence rate of $\{\mathbf{x}_k\}_k$ between gradient descent and CR.

| KŁ parameter | gradient descent | cubic-regularization |
|:---:|:---:|:---:|
| $\theta = 1$ | finite-step | finite-step |
| $\theta \in [\frac{1}{2}, 1)$ | linear | super-linear |
| $\theta \in [\frac{1}{3}, \frac{1}{2})$ | sub-linear | (super)-linear |
| $\theta \in (0, \frac{1}{3})$ | sub-linear $\mathcal{O}(k^{-\frac{\theta}{1-2\theta}})$ | sub-linear $\mathcal{O}(k^{-\frac{2\theta}{1-3\theta}})$ |

## 4.3 Extension to Inexact Cubic Regularization

All our convergence results for CR in previous sections are based on the algorithm dynamics Table 1 and the KŁ property of the objective function. In fact, such dynamics of CR has been shown to be satisfied by other inexact variants of CR Cartis et al. (2011b,a); Kohler and Lucchi (2017); Wang et al. (2018) with different constant terms. These inexact variants of CR updates the variable by solving the cubic subproblem in eq. (1) with the inexact gradient $\nabla \widehat{f}(\mathbf{x}_k)$ and inexact Hessian $\nabla^2 \widehat{f}(\mathbf{x}_k)$ that satisfy the following inexact criterion

$$\|\nabla \widehat{f}(\mathbf{x}_k) - \nabla f(\mathbf{x}_k)\| \leq c_1 \|\mathbf{x}_{k+1} - \mathbf{x}_k\|^2, \tag{5}$$

$$\left\|\left(\nabla^2 \widehat{f}(\mathbf{x}_k) - \nabla^2 f(\mathbf{x}_k)\right)(\mathbf{x}_{k+1} - \mathbf{x}_k)\right\| \leq c_2 \|\mathbf{x}_{k+1} - \mathbf{x}_k\|^2, \tag{6}$$

where $c_1, c_2$ are positive constants. Such inexact criterion can reduce the computational complexity of CR for solving finite-sum problems and can be realized via various types of subsampling schemes Kohler and Lucchi (2017); Wang et al. (2018).

Since the above inexact-CR also satisfies the dynamics in Table 1 (with different constant terms), all our convergence results for CR can be directly applied to inexact CR. Then, we obtain the following corollary.

**Corollary 2** (Inexact-CR). *Let Assumption 1 hold and assume that problem* (P) *satisfies the KŁ property. Then, the sequences* $\{\mu(\mathbf{x}_k)\}_k, \{f(\mathbf{x}_k)\}_k, \{\mathbf{x}_k\}_k$ *generated by the inexact-CR satisfy respectively the results in Theorems 1, 2, 3 and 4.*

## 5  Convergence Rate of CR under KŁ Error Bound

In Yue et al. (2018), it was shown that the gradient dominance condition (i.e., eq. (3) with $\theta = \frac{1}{2}$) implies the following local error bound, which further leads to the quadratic convergence of CR.

**Definition 2.** *Denote $\Omega$ as the set of second-order stationary points of $f$. Then, $f$ is said to satisfy the local error bound condition if there exists $\kappa, \rho > 0$ such that*

$$\mathrm{dist}_\Omega(\mathbf{x}) \le \kappa \|\nabla f(\mathbf{x})\|, \quad \forall \, \mathrm{dist}_\Omega(\mathbf{x}) \le \rho. \tag{7}$$

As the KŁ property generalizes the gradient dominance condition, it implies a much more general spectrum of the geometry that includes the error bound in Yue et al. (2018) as a special case. Next, we first show that the KŁ property implies the KŁ -error bound, and then exploit such an error bound to establish the convergence of CR.

**Proposition 1.** *Denote $\Omega$ as the set of second-order stationary points of $f$. Let Assumption 1 hold and assume that $f$ satisfies the KŁ property. Then, there exist $\kappa, \varepsilon, \lambda > 0$ such that for all $\mathbf{x} \in \{\mathbf{z} \in \mathbb{R}^d : \mathrm{dist}_\Omega(\mathbf{z}) < \varepsilon, f_\Omega < f(\mathbf{z}) < f_\Omega + \lambda\}$, the following property holds.*

$$(\textit{KŁ -error bound}) \quad \mathrm{dist}_\Omega(\mathbf{x}) \le \kappa \|\nabla f(\mathbf{x})\|^{\frac{\theta}{1-\theta}}. \tag{8}$$

We refer to the condition in eq. (8) as the KŁ -error bound, which generalizes the original error bound in eq. (7) under the KŁ property. In particular, the KŁ -error bound reduces to the error bound in the special case $\theta = \frac{1}{2}$. By exploiting the KŁ error bound, we obtain the following convergence result regarding $\mathrm{dist}_\Omega(\mathbf{x}_k)$.

**Proposition 2.** *Denote $\Omega$ as the set of second-order stationary points of $f$. Let Assumption 1 hold and assume that problem* (P) *satisfies the KŁ property. Then, there exists a sufficiently large $k_0 \in \mathbb{N}$ such that for all $k \ge k_0$ the sequence $\{\mathrm{dist}_\Omega(\mathbf{x}_k)\}_k$ generated by CR satisfies*

*1. If $\theta = 1$, then $\mathrm{dist}_\Omega(\mathbf{x}_k) \to 0$ within finite number of iterations;*

*2. If $\theta \in (\frac{1}{3}, 1)$, then $\mathrm{dist}_\Omega(\mathbf{x}_k) \to 0$ super-linearly as $\mathrm{dist}_\Omega(\mathbf{x}_k) \le \Theta\left(\exp\left(-\left(\frac{2\theta}{1-\theta}\right)^{k-k_0}\right)\right)$;*

*3. If $\theta = \frac{1}{3}$, then $\mathrm{dist}_\Omega(\mathbf{x}_k) \to 0$ linearly as $\mathrm{dist}_\Omega(\mathbf{x}_k) \le \Theta\left(\exp\left(-(k-k_0)\right)\right)$;*

*4. If $\theta \in (0, \frac{1}{3})$, then $\mathrm{dist}_\Omega(\mathbf{x}_k) \to 0$ sub-linearly as $\mathrm{dist}_\Omega(\mathbf{x}_k) \le \Theta\left((k-k_0)^{-\frac{2\theta}{1-3\theta}}\right)$.*

We note that Proposition 2 characterizes the convergence rate of the point-to-set distance $\mathrm{dist}_\Omega(\mathbf{x}_k)$, which is different from the convergence rate of the point-to-point distance $\|\mathbf{x}_k - \bar{\mathbf{x}}\|$ established in Theorem 4. Also, the convergence rate results in Proposition 2 generalizes the quadratic convergence result in Yue et al. (2018) that corresponds to the case $\theta = \frac{1}{2}$.

## 6  Conclusion

In this paper, we explore the asymptotic convergence rates of the CR algorithm under the KŁ property of the nonconvex objective function, and establish the convergence rates of function value gap, iterate distance and second-order stationary gap for CR. Our results show that the convergence behavior of CR ranges from sub-linear convergence to super-linear convergence depending on the parameter of the underlying KŁ geometry, and the obtained convergence rates are order-wise improved compared to those of first-order algorithms under the KŁ property. As a future direction, it is interesting to study the convergence of other computationally efficient variants of the CR algorithm such as stochastic variance-reduced CR under the KŁ property in nonconvex optimization.

## Acknowledgments

This work was supported in part by U.S. National Science Foundation under the grants CCF-1761506 and ECCS-1818904.

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
