[Supplementary Material]

# Supplementary Materials

The proof of Theorem 1 is based on the results in other theorems. Thus, we postpone its proof to the end of the supplementary material.

## Proof of Theorem 2

**Theorem 2.** *Let Assumption 1 hold and assume problem* (P) *satisfies the KŁ property associated with parameter $\theta \in (0, 1]$. Then, there exists a sufficiently large $k_0 \in \mathbb{N}$ such that for all $k \geq k_0$ the sequence $\{f(\mathbf{x}_k)\}_k$ generated by CR satisfies*

1. *If $\theta = 1$, then $f(\mathbf{x}_k) \downarrow \bar{f}$ within finite number of iterations;*
2. *If $\theta \in (\frac{1}{3}, 1)$, then $f(\mathbf{x}_k) \downarrow \bar{f}$ super-linearly as $f(\mathbf{x}_{k+1}) - \bar{f} \leq \Theta\Big( \exp\Big( - \big(\frac{2}{3(1-\theta)}\big)^{k-k_0} \big)\Big)$;*
3. *If $\theta = \frac{1}{3}$, then $f(\mathbf{x}_k) \downarrow \bar{f}$ linearly as $f(\mathbf{x}_{k+1}) - \bar{f} \leq \Theta\Big( \exp\big( - (k - k_0) \big)\Big)$;*
4. *If $\theta \in (0, \frac{1}{3})$, then $f(\mathbf{x}_k) \downarrow \bar{f}$ sub-linearly as $f(\mathbf{x}_{k+1}) - \bar{f} \leq \Theta\Big((k - k_0)^{-\frac{2}{1-3\theta}}\Big)$.*

*Proof.* We first recall the following fundamental result proved in Nesterov and Polyak (2006), which serves as a convenient reference.

**Theorem 5** (Theorem 2, Nesterov and Polyak (2006)). *Let Assumption 1 hold. Then, the sequence $\{\mathbf{x}_k\}_k$ generated by CR satisfies*

1. *The set of limit points $\omega(\mathbf{x}_0)$ of $\{\mathbf{x}_k\}_k$ is nonempty and compact, all of which are second-order stationary points;*
2. *The sequence $\{f(\mathbf{x}_k)\}_k$ decreases to a finite limit $\bar{f}$, which is the constant function value evaluated on the set $\omega(\mathbf{x}_0)$.*

From the results of Theorem 5 we conclude that $\text{dist}_{\omega(\mathbf{x}_0)}(\mathbf{x}_k) \to 0$, $f(\mathbf{x}_k) \downarrow \bar{f}$ and $\omega(\mathbf{x}_0)$ is a compact set on which the function value is the constant $\bar{f}$. Then, it is clear that for any fixed $\epsilon > 0, \lambda > 0$ and all $k \geq k_0$ with $k_0$ being sufficiently large, $\mathbf{x}_k \in \{\mathbf{x} : \text{dist}_{\omega(\mathbf{x}_0)}(\mathbf{x}) < \varepsilon, \bar{f} < f(\mathbf{x}) < \bar{f} + \lambda\}$. Hence, all the conditions of the KŁ property in Definition 1 are satisfied, and we can exploit the KŁ inequality in eq. (2).

Denote $r_k := f(\mathbf{x}_k) - \bar{f}$. For all $k \geq k_0$ we obtain that

$$r_k \overset{(i)}{\leq} C\|\nabla f(\mathbf{x}_k)\|^{\frac{1}{1-\theta}} \overset{(ii)}{\leq} C\|\mathbf{x}_k - \mathbf{x}_{k-1}\|^{\frac{2}{1-\theta}} \overset{(iii)}{\leq} C(r_{k-1} - r_k)^{\frac{2}{3(1-\theta)}}, \tag{9}$$

where (i) follows from the KŁ property in eq. (3), (ii) and (iii) follow from the dynamics of CR in Table 1 and we have absorbed all constants into $C$. Define $\delta_k = r_k C^{\frac{3(1-\theta)}{3\theta-1}}$, then the above inequality can be rewritten as

$$\delta_{k-1} - \delta_k \geq \delta_k^{\frac{3(1-\theta)}{2}}, \quad \forall k \geq k_0. \tag{10}$$

Next, we discuss the convergence rate of $\delta_k$ under different regimes of $\theta$.

**Case 1:** $\theta = 1$.

In this case, the KŁ property in eq. (2) satisfies $\varphi'(t) = c$ and implies that $\|\nabla f(\mathbf{x}_k)\| \geq \frac{1}{c}$ for some constant $c > 0$. On the other hand, by the dynamics of CR in Table 1, we obtain that

$$f(\mathbf{x}_{k+1}) \leq f(\mathbf{x}_k) - \frac{M}{12}\|\mathbf{x}_{k+1} - \mathbf{x}_k\|^3 \leq f(\mathbf{x}_k) - \frac{M}{12}(\frac{2}{L+M})^{\frac{3}{2}}\|\nabla f(\mathbf{x}_k)\|^{\frac{3}{2}}. \tag{11}$$

Combining these two facts yields the conclusion that for all $k \geq k_0$

$$f(\mathbf{x}_{k+1}) \leq f(\mathbf{x}_k) - C$$

for some constant $C > 0$. Then, we conclude that $f(\mathbf{x}_k) \downarrow -\infty$, which contradicts the fact that $f(\mathbf{x}_k) \downarrow \bar{f} > -\infty$ (since $f$ is bounded below). Hence, we must have $f(\mathbf{x}_k) \equiv \bar{f}$ for all sufficiently large $k$.

**Case 2:** $\theta \in (\frac{1}{3}, 1)$.

In this case $0 < \frac{3(1-\theta)}{2} < 1$. Since $\delta_k \to 0$ as $r_k \to 0$, $\delta_k^{\frac{3(1-\theta)}{2}}$ is order-wise larger than $\delta_k$ for all sufficiently large $k$. Hence, for all sufficiently large $k$, eq. (10) reduces to

$$\delta_{k-1} \geq \delta_k^{\frac{3(1-\theta)}{2}}. \tag{12}$$

It follows that $\delta_k \downarrow 0$ super-linearly as $\delta_k \leq \delta_{k-1}^{\frac{2}{3(1-\theta)}}$. Since $\delta_k = r_k C^{\frac{2}{1-3\theta}}$, we conclude that $r_k \downarrow 0$ super-linearly as $r_k \leq C_1 r_{k-1}^{\frac{2}{3(1-\theta)}}$ for some constant $C_1 > 0$. By letting $k_0$ be sufficiently large so that $r_{k_0}$ is sufficiently small, we obtain that

$$r_k \leq C_1 r_{k-1}^{\frac{2}{3(1-\theta)}} \leq C_1^{k-k_0} r_{k_0}^{(\frac{2}{3(1-\theta)})^{k-k_0}} = \Theta\left( \exp\left( -\left(\frac{2}{3(1-\theta)}\right)^{k-k_0} \right) \right). \tag{13}$$

**Case 3:** $\theta = \frac{1}{3}$.

In this case $\frac{3(1-\theta)}{2} = 1$, and eq. (9) reduces to $r_k \leq C(r_{k-1} - r_k)$, i.e., $r_k \downarrow 0$ linearly as $r_k \leq \frac{C}{1+C} r_{k-1}$ for some constant $C > 0$. Thus, we obtain that for all $k \geq k_0$

$$r_k \leq \left(\frac{C}{1+C}\right)^{k-k_0} r_{k_0} = \Theta\left( \exp\left( -(k-k_0) \right) \right). \tag{14}$$

**Case 4:** $\theta \in (0, \frac{1}{3})$.

In this case, $1 < \frac{3(1-\theta)}{2} < \frac{3}{2}$ and $-\frac{1}{2} < \frac{3\theta-1}{2} < 0$. Since $\delta_k \downarrow 0$, we conclude that for all $k \geq k_0$

$$\delta_{k-1}^{-\frac{3(1-\theta)}{2}} < \delta_k^{-\frac{3(1-\theta)}{2}}, \quad \delta_{k-1}^{\frac{3\theta-1}{2}} < \delta_k^{\frac{3\theta-1}{2}}. \tag{15}$$

Define an auxiliary function $\phi(t) := \frac{2}{1-3\theta} t^{\frac{3\theta-1}{2}}$ so that $\phi'(t) = -t^{\frac{3(\theta-1)}{2}}$. We next consider two cases. First, suppose that $\delta_k^{\frac{3(\theta-1)}{2}} \leq 2\delta_{k-1}^{\frac{3(\theta-1)}{2}}$. Then for all $k \geq k_0$

$$\phi(\delta_k) - \phi(\delta_{k-1}) = \int_{\delta_{k-1}}^{\delta_k} \phi'(t)dt = \int_{\delta_k}^{\delta_{k-1}} t^{\frac{3(\theta-1)}{2}} dt \geq (\delta_{k-1} - \delta_k)\delta_{k-1}^{\frac{3(\theta-1)}{2}} \tag{16}$$

$$\overset{(i)}{\geq} \frac{1}{2}(\delta_{k-1} - \delta_k)\delta_k^{\frac{3(\theta-1)}{2}} \overset{(ii)}{\geq} \frac{1}{2}, \tag{17}$$

where (i) utilizes the assumption and (ii) uses eq. (10).

Second, suppose that $\delta_k^{\frac{3(\theta-1)}{2}} \geq 2\delta_{k-1}^{\frac{3(\theta-1)}{2}}$. Then $\delta_k^{\frac{3\theta-1}{2}} \geq 2^{\frac{3\theta-1}{3(\theta-1)}}\delta_{k-1}^{\frac{3\theta-1}{2}}$, which further leads to

$$\phi(\delta_k) - \phi(\delta_{k-1}) = \frac{2}{1-3\theta}(\delta_k^{\frac{3\theta-1}{2}} - \delta_{k-1}^{\frac{3\theta-1}{2}}) \geq \frac{2}{1-3\theta}(2^{\frac{3\theta-1}{3(\theta-1)}} - 1)\delta_{k-1}^{\frac{3\theta-1}{2}} \tag{18}$$

$$\geq \frac{2}{1-3\theta}(2^{\frac{3\theta-1}{3(\theta-1)}} - 1)\delta_{k_0}^{\frac{3\theta-1}{2}}. \tag{19}$$

Combining the above two cases and defining $C := \min\{\frac{1}{2}, \frac{2}{1-3\theta}(2^{\frac{3\theta-1}{3(\theta-1)}} - 1)\delta_{k_0}^{\frac{3\theta-1}{2}}\}$, we conclude that for all $k \geq k_0$

$$\phi(\delta_k) - \phi(\delta_{k-1}) \geq C, \tag{20}$$

which further implies that

$$\phi(\delta_k) \geq \sum_{i=k_0+1}^{k} \phi(\delta_i) - \phi(\delta_{i-1}) \geq C(k-k_0). \tag{21}$$

Substituting the form of $\phi$ into the above inequality and simplifying the expression yields $\delta_k \leq \left(\frac{2}{C(1-3\theta)(k-k_0)}\right)^{\frac{2}{1-3\theta}}$. It follows that $r_k \leq \left(\frac{C_3}{k-k_0}\right)^{\frac{2}{1-3\theta}}$ for some $C_3 > 0$.

$\square$

## Proof of Theorem 3

**Theorem 3.** *Let Assumption 1 hold and assume that problem* (P) *satisfies the KŁ property. Then, the sequence* $\{\mathbf{x}_k\}_k$ *generated by CR satisfies*

$$\sum_{k=0}^{\infty} \|\mathbf{x}_{k+1} - \mathbf{x}_k\| < +\infty. \tag{4}$$

*Proof.* Recall the definition that $r_k := f(\mathbf{x}_k) - \bar{f}$, where $\bar{f}$ is the finite limit of $\{f(\mathbf{x}_k)\}_k$. Also, recall that $k_0 \in \mathbb{N}$ is a sufficiently large integer. Then, for all $k \geq k_0$, the KŁ property implies that

$$\varphi'(r_k) \geq \frac{1}{\|\nabla f(\mathbf{x}_k)\|} \geq \frac{2}{(L+M)\|\mathbf{x}_k - \mathbf{x}_{k-1}\|^2}, \tag{22}$$

where the last inequality uses the dynamics of CR in Table 1. Note that $\varphi(t) = \frac{c}{\theta} t^{\theta}$ is concave for $\theta \in (0, 1]$. Then, by concavity we obtain that

$$\varphi(r_k) - \varphi(r_{k+1}) \geq \varphi'(r_k)(r_k - r_{k+1}) \geq \frac{M}{6(L+M)} \frac{\|\mathbf{x}_{k+1} - \mathbf{x}_k\|^3}{\|\mathbf{x}_k - \mathbf{x}_{k-1}\|^2}, \tag{23}$$

where the last inequality uses eq. (22) and the dynamics of CR in Table 1. Rearranging the above inequality, taking cubic root and summing over $k = k_0, \ldots, n$ yield that (all constants are absorbed in $C$)

$$\sum_{k=k_0}^{n} \|\mathbf{x}_{k+1} - \mathbf{x}_k\| \leq C \sum_{k=k_0}^{n} (\varphi(r_k) - \varphi(r_{k+1}))^{\frac{1}{3}} \|\mathbf{x}_k - \mathbf{x}_{k-1}\|^{\frac{2}{3}} \tag{24}$$

$$\overset{(i)}{\leq} C \left[ \sum_{k=k_0}^{n} (\varphi(r_k) - \varphi(r_{k+1})) \right]^{\frac{1}{3}} \left[ \sum_{k=k_0}^{n} \|\mathbf{x}_k - \mathbf{x}_{k-1}\| \right]^{\frac{2}{3}} \tag{25}$$

$$\overset{(ii)}{\leq} C \left[\varphi(r_{k_0})\right]^{\frac{1}{3}} \left[ \sum_{k=k_0}^{n} \|\mathbf{x}_{k+1} - \mathbf{x}_k\| + \|\mathbf{x}_{k_0} - \mathbf{x}_{k_0-1}\| \right]^{\frac{2}{3}}, \tag{26}$$

where (i) applies the Hölder's inequality and (ii) uses the fact that $\varphi \geq 0$. Clearly, we must have $\lim_{n \to \infty} \sum_{k=k_0}^{n} \|\mathbf{x}_{k+1} - \mathbf{x}_k\| < +\infty$, because otherwise the above inequality cannot hold for all $n$ sufficiently large. We then conclude that

$$\sum_{k=k_0}^{\infty} \|\mathbf{x}_{k+1} - \mathbf{x}_k\| < +\infty,$$

and the desired result follows because $k_0$ is a fixed number.

$\square$

## Proof of Theorem 4

**Theorem 4.** *Let Assumption 1 hold and assume that problem* (P) *satisfies the KŁ property. Then, there exists a sufficiently large* $k_0 \in \mathbb{N}$ *such that for all* $k \geq k_0$ *the sequence* $\{\mathbf{x}_k\}_k$ *generated by CR satisfies*

1. *If* $\theta = 1$, *then* $\mathbf{x}_k \to \bar{\mathbf{x}}$ *within finite number of iterations;*
2. *If* $\theta \in (\frac{1}{3}, 1)$, *then* $\mathbf{x}_k \to \bar{\mathbf{x}}$ *super-linearly as* $\|\mathbf{x}_{k+1} - \bar{\mathbf{x}}\| \leq \Theta\left( \exp\left( -\left(\frac{2\theta}{3(1-\theta)} + \frac{2}{3}\right)^{k-k_0}\right)\right)$;
3. *If* $\theta = \frac{1}{3}$, *then* $\mathbf{x}_k \to \bar{\mathbf{x}}$ *linearly as* $\|\mathbf{x}_{k+1} - \bar{\mathbf{x}}\| \leq \Theta\left( \exp\left( -(k - k_0)\right)\right)$;
4. *If* $\theta \in (0, \frac{1}{3})$, *then* $\mathbf{x}_k \to \bar{\mathbf{x}}$ *sub-linearly as* $\|\mathbf{x}_{k+1} - \bar{\mathbf{x}}\| \leq \Theta\left((k - k_0)^{-\frac{2\theta}{1-3\theta}}\right)$.

*Proof.* We prove the theorem case by case.

**Case 1: $\theta = 1$.**

We have shown in case 1 of Theorem 2 that $f(\mathbf{x}_k) \downarrow \bar{f}$ within finite number of iterations, i.e., $f(\mathbf{x}_{k+1}) - f(\mathbf{x}_k) = 0$ for all $k \geq k_0$. Based on this observation, the dynamics of CR in Table 1 further implies that for all $k \geq k_0$

$$0 = f(\mathbf{x}_{k+1}) - f(\mathbf{x}_k) \leq -\frac{M}{12}\|\mathbf{x}_{k+1} - \mathbf{x}_k\|^3 \leq 0. \tag{27}$$

Hence, we conclude that $\mathbf{x}_{k+1} = \mathbf{x}_k$ for all $k \geq k_0$, i.e., $\mathbf{x}_k$ converges within finite number of iterations. Since Theorem 3 shows that $\mathbf{x}_k$ converges to some $\bar{\mathbf{x}}$, the desired conclusion follows.

**Case 2: $\theta \in (\frac{1}{3}, 1)$.**

Denote $\Delta_k := \sum_{i=k}^{\infty} \|\mathbf{x}_{i+1} - \mathbf{x}_i\|$. Note that Theorem 3 shows that $\mathbf{x}_k \to \bar{\mathbf{x}}$. Thus, we have $\|\mathbf{x}_k - \bar{\mathbf{x}}\| \leq \Delta_k$. Next, we derive the convergence rate of $\Delta_k$.

By Theorem 3, $\lim_{n\to\infty} \sum_{i=k}^{n} \|\mathbf{x}_{i+1} - \mathbf{x}_i\|$ exists for all $k$. Then, we can let $n \to \infty$ in eq. (26) and obtain that for all $k \geq k_0$

$$\Delta_k \leq C[\varphi(r_k)]^{\frac{1}{3}} \Delta_{k-1}^{\frac{2}{3}} \leq Cr_k^{\frac{\theta}{3}} \Delta_{k-1}^{\frac{2}{3}} \overset{(i)}{\leq} C(\Delta_{k-1} - \Delta_k)^{\frac{2\theta}{3(1-\theta)}} \Delta_{k-1}^{\frac{2}{3}} \leq C\Delta_{k-1}^{\frac{2\theta}{3(1-\theta)} + \frac{2}{3}}, \tag{28}$$

where $C$ denotes a universal constant that may vary from line to line, and (i) uses the KŁ property and the dynamics of CR, i.e., $r_k \leq C\|\nabla f(\mathbf{x}_k)\|^{\frac{1}{1-\theta}} \leq C\|\mathbf{x}_k - \mathbf{x}_{k-1}\|^{\frac{2}{1-\theta}}$. Note that in this case we have $\frac{2\theta}{3(1-\theta)} + \frac{2}{3} > 1$, and hence the above inequality implies that $\Delta_k$ converges to zero super-linearly as

$$\Delta_k \leq C^{k-k_0} \Delta_{k_0}^{(\frac{2\theta}{3(1-\theta)} + \frac{2}{3})^{k-k_0}} = \Theta\left(\exp\left(-\left(\frac{2\theta}{3(1-\theta)} + \frac{2}{3}\right)^{k-k_0}\right)\right). \tag{29}$$

Since $\|\mathbf{x}_k - \bar{\mathbf{x}}\| \leq \Delta_k$, it follows that $\|\mathbf{x}_k - \bar{\mathbf{x}}\|$ converges to zero super-linearly as desired.

**Cases 3 & 4.**

We first derive another estimate on $\Delta_k$ that generally holds for both cases 3 and 4, and then separately consider cases 3 and 4, respectively.

Fix $\gamma \in (0, 1)$ and consider $k \geq k_0$. Suppose that $\|\mathbf{x}_{k+1} - \mathbf{x}_k\| \geq \gamma\|\mathbf{x}_k - \mathbf{x}_{k-1}\|$, then eq. (23) can be rewritten as

$$\|\mathbf{x}_{k+1} - \mathbf{x}_k\| \leq \frac{C}{\gamma^2}(\varphi(r_k) - \varphi(r_{k+1})) \tag{30}$$

for some constant $C > 0$. Otherwise, we have $\|\mathbf{x}_{k+1} - \mathbf{x}_k\| \leq \gamma\|\mathbf{x}_k - \mathbf{x}_{k-1}\|$. Combing these two inequalities yields that

$$\|\mathbf{x}_{k+1} - \mathbf{x}_k\| \leq \gamma\|\mathbf{x}_k - \mathbf{x}_{k-1}\| + \frac{C}{\gamma^2}(\varphi(r_k) - \varphi(r_{k+1})). \tag{31}$$

Summing the above inequality over $k = k_0, \ldots, n$ yields that

$$\sum_{k=k_0}^{n} \|\mathbf{x}_{k+1} - \mathbf{x}_k\| \leq \gamma \sum_{k=k_0}^{n} \|\mathbf{x}_k - \mathbf{x}_{k-1}\| + \frac{C}{\gamma^2}(\varphi(r_{k_0}) - \varphi(r_{n+1})) \tag{32}$$

$$\leq \gamma\left[\sum_{k=k_0}^{n} \|\mathbf{x}_{k+1} - \mathbf{x}_k\| + \|\mathbf{x}_{k_0} - \mathbf{x}_{k_0-1}\|\right] + \frac{C}{\gamma^2}\varphi(r_{k_0}). \tag{33}$$

Rearranging the above inequality yields that

$$\sum_{k=k_0}^{n} \|\mathbf{x}_{k+1} - \mathbf{x}_k\| \leq \frac{\gamma}{1-\gamma}\|\mathbf{x}_{k_0} - \mathbf{x}_{k_0-1}\| + \frac{C}{\gamma^2(1-\gamma)}\varphi(r_{k_0}). \tag{34}$$

Recall $\Delta_k := \sum_{i=k}^{\infty} \|\mathbf{x}_{i+1} - \mathbf{x}_i\| < +\infty$. Letting $n \to \infty$ in the above inequality yields that for all sufficiently large $k$

$$\Delta_k \leq \frac{\gamma}{1-\gamma}(\Delta_{k-1} - \Delta_k) + \frac{C}{\gamma^2(1-\gamma)\theta}r_k^{\theta} \tag{35}$$

$$\overset{(i)}{\leq} \frac{\gamma}{1-\gamma}(\Delta_{k-1} - \Delta_k) + \frac{C}{\gamma^2(1-\gamma)\theta}\|\mathbf{x}_k - \mathbf{x}_{k-1}\|^{\frac{2\theta}{1-\theta}} \tag{36}$$

$$\leq \frac{\gamma}{1-\gamma}(\Delta_{k-1} - \Delta_k) + \frac{C}{\gamma^2(1-\gamma)\theta}(\Delta_{k-1} - \Delta_k)^{\frac{2\theta}{1-\theta}}, \tag{37}$$

where (i) uses the KŁ property and the dynamics of CR, i.e., $r_k \leq C\|\nabla f(\mathbf{x}_k)\|^{\frac{1}{1-\theta}} \leq C\|\mathbf{x}_k - \mathbf{x}_{k-1}\|^{\frac{2}{1-\theta}}$.

**Case 3:** $\theta = \frac{1}{3}$. In this case, $\frac{2\theta}{1-\theta} = 1$ and eq. (37) implies that $\Delta_k \leq C(\Delta_{k-1} - \Delta_k)$ for all sufficiently large $k$, i.e., $\Delta_k$ converges to zero linearly as $\Delta_k \leq (\frac{C}{1+C})^{k-k_0}\Delta_{k_0}$. The desired result follows since $\|\mathbf{x}_k - \bar{\mathbf{x}}\| \leq \Delta_k$.

**Case 4:** $\theta \in (0, \frac{1}{3})$. In this case, $0 < \frac{2\theta}{1-\theta} < 1$ and eq. (37) can be asymptotically rewritten as $\Delta_k \leq \frac{C}{\gamma^2(1-\gamma)\theta}(\Delta_{k-1} - \Delta_k)^{\frac{2\theta}{1-\theta}}$. This further implies that

$$\Delta_k^{\frac{1-\theta}{2\theta}} \leq C(\Delta_{k-1} - \Delta_k) \tag{38}$$

for some constant $C > 0$. Define $h(t) = t^{-\frac{1-\theta}{2\theta}}$ and fix $\beta > 1$. Suppose first that $h(\Delta_k) \leq \beta h(\Delta_{k-1})$. Then the above inequality implies that

$$1 \leq C\frac{\Delta_{k-1} - \Delta_k}{\Delta_k^{\frac{1-\theta}{2\theta}}} = C(\Delta_{k-1} - \Delta_k)h(\Delta_k) \leq C\beta(\Delta_{k-1} - \Delta_k)h(\Delta_{k-1}) \tag{39}$$

$$\leq C\beta \int_{\Delta_k}^{\Delta_{k-1}} h(t)dt = C\beta\frac{2\theta}{3\theta-1}(\Delta_{k-1}^{\frac{3\theta-1}{2\theta}} - \Delta_k^{\frac{3\theta-1}{2\theta}}). \tag{40}$$

Set $\mu := \frac{1-3\theta}{2C\beta\theta} > 0, \nu := \frac{3\theta-1}{2\theta} < 0$. Then the above inequality can be rewritten as

$$\Delta_k^{\nu} - \Delta_{k-1}^{\nu} \geq \mu. \tag{41}$$

Now suppose $h(\Delta_k) > \beta h(\Delta_{k-1})$, which implies that $\Delta_k < q\Delta_{k-1}$ with $q = \beta^{-\frac{2\theta}{1-\theta}} \in (0,1)$. Then, we conclude that $\Delta_k^{\nu} \geq q^{\nu}\Delta_{k-1}^{\nu}$ and hence $\Delta_k^{\nu} - \Delta_{k-1}^{\nu} \geq (q^{\nu} - 1)\Delta_{k-1}^{\nu}$. Since $q^{\nu} - 1 > 0$ and $\Delta_{k-1}^{\nu} \to +\infty$, there must exist $\bar{\mu} > 0$ such that $(q^{\nu} - 1)\Delta_{k-1}^{\nu} \geq \bar{\mu}$ for all sufficiently large $k$. Thus, we conclude that $\Delta_k^{\nu} - \Delta_{k-1}^{\nu} \geq \bar{\mu}$. Combining two cases, we obtain that for all sufficiently large $k$,

$$\Delta_k^{\nu} - \Delta_{k-1}^{\nu} \geq \min\{\mu, \bar{\mu}\}. \tag{42}$$

Telescoping the above inequality over $k = k_0, \ldots, k$ yields that

$$\Delta_k \leq [\Delta_{k_0}^{\nu} + \min\{\mu, \bar{\mu}\}(k - k_0)]^{\frac{1}{\nu}} \leq \left(\frac{C}{k-k_0}\right)^{\frac{2\theta}{1-3\theta}}, \tag{43}$$

where $C$ is a certain positive constant. The desired result then follows from the fact that $\|\mathbf{x}_k - \bar{\mathbf{x}}\| \leq \Delta_k$. $\qquad\square$

## Proof of Proposition 1

**Proposition 1.** *Denote $\Omega$ as the set of second-order stationary points of $f$. Let Assumption 1 hold and assume that $f$ satisfies the KŁ property. Then, there exist $\kappa, \varepsilon, \lambda > 0$ such that for all $\mathbf{x} \in \{\mathbf{z} \in \mathbb{R}^d : \mathrm{dist}_\Omega(\mathbf{z}) < \varepsilon, f_\Omega < f(\mathbf{z}) < f_\Omega + \lambda\}$, the following property holds.*

$$(\text{KŁ -error bound}) \quad \mathrm{dist}_\Omega(\mathbf{x}) \leq \kappa\|\nabla f(\mathbf{x})\|^{\frac{\theta}{1-\theta}}. \tag{8}$$

*Proof.* The proof idea follows from that in Yue et al. (2018). Consider any $\mathbf{x} \in \Omega^c \cap \{\mathbf{x} \in \mathbb{R}^d : \text{dist}_\Omega(\mathbf{x}) < \varepsilon, f_\Omega < f(\mathbf{x}) < f_\Omega + \lambda\}$, and consider the following differential equation

$$\mathbf{u}(0) = \mathbf{x}, \quad \dot{\mathbf{u}}(t) = -\nabla f(\mathbf{u}(t)), \quad \forall t > 0. \tag{44}$$

As $\nabla f$ is continuously differentiable, it is Lipschitz on every compact set. Thus, by the Picard-Lindelöf theorem (Hartman, 2002, Theorem II.1.1), there exists $\nu > 0$ such that eq. (44) has a unique solution $\mathbf{u_x}(t)$ over the interval $[0, \nu]$. Define $\Delta(t) := f(\mathbf{u_x}(t)) - f_\Omega$. Note that $\Delta(t) > 0$ for $t \in [0, \nu]$, as otherwise there exists $\hat{t} \in [0, \nu]$ such that $\mathbf{u_x}(\hat{t}) \in \Omega$ and hence $\mathbf{u_x} \equiv \mathbf{u_x}(\hat{t}) \in \Omega$ is the unique solution to eq. (44). This contradicts the fact that $\mathbf{u}(0) \in \Omega^c$.

Using eq. (44) and the chain rule, we obtain that for all $t \in [0, \nu]$

$$\dot{\Delta}(t) = \langle \nabla f(\mathbf{u_x}(t)), \dot{\mathbf{u}}_\mathbf{x}(t) \rangle = -\|\nabla f(\mathbf{u_x}(t))\| \|\dot{\mathbf{u}}_\mathbf{x}(t)\|. \tag{45}$$

Applying the KŁ property in eq. (3) to the above equation yields that

$$\dot{\Delta}(t) \leq -\left(\frac{\Delta(t)}{C}\right)^{1-\theta} \|\dot{\mathbf{u}}_\mathbf{x}(t)\|, \tag{46}$$

where $C > 0$ is a certain universal constant. Since $\Delta(t) > 0$, eq. (46) can be rewritten as

$$\|\dot{\mathbf{u}}_\mathbf{x}(t)\| \leq -\frac{C^{1-\theta}}{\theta}(\Delta(t)^\theta)'. \tag{47}$$

Based on the above inequality, for any $0 \leq a < b < \nu$ we obtain that

$$\|\mathbf{u_x}(b) - \mathbf{u_x}(a)\| = \|\int_a^b \dot{\mathbf{u}}_\mathbf{x}(t)dt\| \leq \int_a^b \|\dot{\mathbf{u}}_\mathbf{x}(t)\| dt$$
$$\leq -\int_a^b \frac{C^{1-\theta}}{\theta}[\Delta(t)^\theta]' dt = \frac{C^{1-\theta}}{\theta}[\Delta(a)^\theta - \Delta(b)^\theta]. \tag{48}$$

In particular, setting $a = 0$ in eq. (48) and noting that $\mathbf{u_x}(0) = \mathbf{x}$, we further obtain that

$$\|\mathbf{u_x}(b) - \mathbf{x}\| \leq \frac{C^{1-\theta}}{\theta}(f(\mathbf{x}) - f_\Omega)^\theta. \tag{49}$$

Next, we show that $\nu = +\infty$. Suppose $\nu < +\infty$, then (Hartman, 2002, Corollary II.3.2) shows that $\|\mathbf{u_x}(t)\| \to +\infty$ as $t \to \nu$. However, eq. (49) implies that

$$\|\mathbf{u_x}(t)\| \leq \|\mathbf{x}\| + \|\mathbf{u_x}(t) - \mathbf{x}\| \leq \|\mathbf{x}\| + \frac{C^{1-\theta}}{\theta}(f(\mathbf{x}) - f_\Omega)^\theta < +\infty,$$

which leads to a contradiction. Thus, $\nu = +\infty$.

Since $\dot{\Delta}(t) \leq 0$, $\Delta(t)$ is non-increasing. Hence, the nonnegative sequence $\{\Delta(t)\}$ has a limit. Then, eq. (48) further implies that $\{\mathbf{u_x}(t)\}$ is a Cauchy sequence and hence has a limit $\mathbf{u_x}(\infty)$. Suppose $\nabla f(\mathbf{u_x}(\infty)) \neq \mathbf{0}$. Then we obtain that $\lim_{t\to\infty} \dot{\Delta}(t) = -\|\nabla f(\mathbf{u_x}(\infty))\|^2 < 0$, which contradicts the fact that $\lim_{t\to\infty} \Delta(t)$ exists. Thus, $\nabla f(\mathbf{u_x}(\infty)) = \mathbf{0}$, and this further implies that $\mathbf{u_x}(\infty) \in \Omega, f(\mathbf{u_x}(\infty)) = f_\Omega$ by the KŁ property in eq. (3). We then conclude that

$$\text{dist}_\Omega(\mathbf{x}) \leq \|\mathbf{x} - \mathbf{u_x}(\infty)\| = \lim_{t\to\infty} \|\mathbf{x} - \mathbf{u_x}(t)\| \leq \frac{C^{1-\theta}}{\theta}(f(\mathbf{x}) - f_\Omega)^\theta. \tag{50}$$

Combining the above inequality with the KŁ property in eq. (3), we obtain the desired KŁ error bound. □

## Proof of Proposition 2

**Proposition 2.** *Denote $\Omega$ as the set of second-order stationary points of $f$. Let Assumption 1 hold and assume that problem* (P) *satisfies the KŁ property. Then, there exists a sufficiently large $k_0 \in \mathbb{N}$ such that for all $k \geq k_0$ the sequence $\{\text{dist}_\Omega(\mathbf{x}_k)\}_k$ generated by CR satisfies*

1. *If $\theta = 1$, then $\mathrm{dist}_\Omega(\mathbf{x}_k) \to 0$ within finite number of iterations;*
2. *If $\theta \in (\frac{1}{3}, 1)$, then $\mathrm{dist}_\Omega(\mathbf{x}_k) \to 0$ super-linearly as $\mathrm{dist}_\Omega(\mathbf{x}_k) \le \Theta\left( \exp\left( - \left(\frac{2\theta}{1-\theta}\right)^{k-k_0} \right) \right)$;*
3. *If $\theta = \frac{1}{3}$, then $\mathrm{dist}_\Omega(\mathbf{x}_k) \to 0$ linearly as $\mathrm{dist}_\Omega(\mathbf{x}_k) \le \Theta\left( \exp\left( - (k - k_0) \right) \right)$;*
4. *If $\theta \in (0, \frac{1}{3})$, then $\mathrm{dist}_\Omega(\mathbf{x}_k) \to 0$ sub-linearly as $\mathrm{dist}_\Omega(\mathbf{x}_k) \le \Theta\left( (k - k_0)^{-\frac{2\theta}{1-3\theta}} \right)$.*

*Proof.* We prove the theorem case by case.

**Case 1: $\theta = 1$.**

We have proved in Theorem 4 that $\mathbf{x}_k \to \bar{\mathbf{x}} \in \Omega$ within finite number of iterations. Since $\mathrm{dist}_\Omega(\mathbf{x}_k) \le \|\mathbf{x}_k - \bar{\mathbf{x}}\|$, we conclude that $\mathrm{dist}_\Omega(\mathbf{x}_k)$ converges to zero within finite number of iterations.

**Case 2: $\theta \in (\frac{1}{3}, 1)$.**

By the KŁ error bound in Proposition 1, we obtain that

$$\mathrm{dist}_\Omega(\mathbf{x}_{k+1}) \le C\|\nabla f(\mathbf{x}_{k+1})\|^{\frac{\theta}{1-\theta}} \le C\|\mathbf{x}_{k+1} - \mathbf{x}_k\|^{\frac{2\theta}{1-\theta}}, \tag{51}$$

where the last inequality uses the dynamics of CR in Table 1. On the other hand, (Yue et al., 2018, Lemma 1) shows that

$$\|\mathbf{x}_{k+1} - \mathbf{x}_k\| \le C\mathrm{dist}_\Omega(\mathbf{x}_k). \tag{52}$$

Combining eq. (51) and eq. (52) yields that

$$\mathrm{dist}_\Omega(\mathbf{x}_{k+1}) \le C\mathrm{dist}_\Omega(\mathbf{x}_k)^{\frac{2\theta}{1-\theta}}. \tag{53}$$

Note that in this case we have $\frac{2\theta}{1-\theta} > 1$. Thus, $\mathrm{dist}_\Omega(\mathbf{x}_k)$ converges to zero super-linearly as desired.

**Cases 3 & 4: $\theta \in (0, \frac{1}{3}]$.**

Note that $\mathrm{dist}_\Omega(\mathbf{x}_k) \le \|\mathbf{x}_k - \bar{\mathbf{x}}\|$. The desired results follow from Cases 3 & 4 in Theorem 4.

$\square$

## Proof of Theorem 1

**Theorem 1.** *Let Assumption 1 hold and assume that problem* (P) *satisfies the KŁ property associated with parameter $\theta \in (0, 1]$. Then, there exists a sufficiently large $k_0 \in \mathbb{N}$ such that for all $k \ge k_0$ the sequence $\{\mu(\mathbf{x}_k)\}_k$ generated by CR satisfies*

1. *If $\theta = 1$, then $\mu(\mathbf{x}_k) \to 0$ within finite number of iterations;*
2. *If $\theta \in (\frac{1}{3}, 1)$, then $\mu(\mathbf{x}_k) \to 0$ super-linearly as $\mu(\mathbf{x}_k) \le \Theta\left( \exp\left( - \left(\frac{2\theta}{1-\theta}\right)^{k-k_0} \right) \right)$;*
3. *If $\theta = \frac{1}{3}$, then $\mu(\mathbf{x}_k) \to 0$ linearly as $\mu(\mathbf{x}_k) \le \Theta\left( \exp\left( - (k - k_0) \right) \right)$;*
4. *If $\theta \in (0, \frac{1}{3})$, then $\mu(\mathbf{x}_k) \to 0$ sub-linearly as $\mu(\mathbf{x}_k) \le \Theta\left( (k - k_0)^{-\frac{2\theta}{1-3\theta}} \right)$.*

*Proof.* By the dynamics of CR in Table 1, we obtain that

$$\|\nabla f(\mathbf{x}_{k+1})\| \le \frac{L + M}{2}\|\mathbf{x}_{k+1} - \mathbf{x}_k\|^2, \tag{54}$$

$$-\lambda_{\min}(\nabla^2 f(\mathbf{x}_{k+1})) \le \frac{2L + M}{2}\|\mathbf{x}_{k+1} - \mathbf{x}_k\|. \tag{55}$$

The above two inequalities imply that $\mu(\mathbf{x}_k) \le \|\mathbf{x}_{k+1} - \mathbf{x}_k\|$. Also, (Yue et al., 2018, Lemma 1) shows that $\|\mathbf{x}_{k+1} - \mathbf{x}_k\| \le C\mathrm{dist}_\Omega(\mathbf{x}_k)$. Then, the desired convergence result for $\mu(\mathbf{x}_k)$ follows from Proposition 2.

$\square$