[Reviews · NeurIPS 2018]

Reviewer 1



############# The authors have addressed my previous comments. ############# In this paper, the authors provided a comprehensive study on the convergence rate of CR under KL condition. Existing work restricted CR to convergence analysis conditioned on special types of geometrical properties of the objective function. Thus, this work is novel and contains significant theoretical contributions. Strengths: 1) This work provided a comprehensive study on the convergence rate of CR under four different optimally measures. 2) Table 1-3 clearly showed the differences with the existing work, and could motivate further work in the field. Weakness: It lacks numerical results to verify the theoretical results proposed in this paper. Comment: 1) Please highlight the technical difficulty of convergence analysis under KL compared to the existing work. 2) $\phi^\prime$ was not defined in Eq. (2)?

Reviewer 2



This paper study the asymptotic convergence rate of cubic regularization algorithm under the Kurdyka-Łojasiewicz (KŁ) property. Comparing with existing works on convergence analysis for CR method --- a sublinear global convergence rate and a local quadratic rate under certain conditions, this paper provides a more fine-grained convergence rate over a broad spectrum of function local geometries characterized by the KŁ property. These results help better understand the local convergence behavior of CR methods. It will be much better to show some concrete examples (even toy examples) for each case of the different KL properties and verify the convergence behavior of CR method.

Reviewer 3



The paper investigates the convergence of different measurement of cubic regularization method for non-convex optimization under KL property. It consists with a list of work on CR methods based on the analysis of Nesterove el.s' work. Since the type of methods can guarantee the convergence to the second-order stationary point, it is quite popular also considering the raising of training neural networks. The paper is well-written, clear-organized and the theorems and proofs are easy to follow. Note that this is a pure theoretical work i.e., without new algorithms and/or numerical experiments. I have the following concerns 1. The results of the paper is based on KL property, and show convergence result in terms of the critical parameter \theta of KL. As a generalized function property, it is not easy to have \theta for a given function and the paper does not provide examples where this \theta can be values other than \theta=1/2 for some machine learning problems. 2. To apply the algorithm, the Lipschitz-Hessian constant and an exact solution for subproblem is needed to derive the second-order stationary point convergence, which makes the algorithm unpractical for large problems. 3. There is no discussion over finite sum problems, which is typical in machine learning field, which I believe NIPS has more focus on such problems. Considering the above, the paper is good but might not suitable for NIPS. I would therefore only vote the paper a weak accept but I'm also open to others suggestions. minors: 1. \bar{x} in Definition 1 is not used and make the definition confusing.